# Chronic Kidney Disease in the Older Adult Patient with Diabetes

**DOI:** 10.3390/jcm13020348

**Published:** 2024-01-08

**Authors:** Raja Ravender, Maria-Eleni Roumelioti, Darren W. Schmidt, Mark L. Unruh, Christos Argyropoulos

**Affiliations:** Division of Nephrology, Department of Internal Medicine, University of New Mexico School of Medicine, MSC 04-2785, Albuquerque, NM 87131, USA; rravender@salud.unm.edu (R.R.); mroumelioti@salud.unm.edu (M.-E.R.); dwschmidt@salud.unm.edu (D.W.S.); mlunruh@salud.unm.edu (M.L.U.)

**Keywords:** diabetes, chronic kidney disease, treatment, elderly, geriatric, dialysis, SGLT2 inhibitors, GLP1 receptor agonists, non-steroidal mineralocorticoid antagonists

## Abstract

Diabetes mellitus (DM) and chronic kidney disease (CKD) are common in middle aged and older adult individuals. DM may accelerate the aging process, and the age-related declines in the estimated glomerular filtration rate (eGFR) can pose a challenge to diagnosing diabetic kidney disease (DKD) using standard diagnostic criteria especially with the absence of severe albuminuria among older adults. In the presence of CKD and DM, older adult patients may need multidisciplinary care due to susceptibility to various health issues, e.g., cognitive decline, auditory or visual impairment, various comorbidities, complex medical regimens, and increased sensitivity to medication adverse effects. As a result, it can be challenging to apply recent therapeutic advancements for the general population to older adults. We review the evidence that the benefits from these newer therapies apply equally to older and younger patients with CKD and diabetes type 2 and propose a comprehensive management. This framework will address nonpharmacological measures and pharmacological management with renin angiotensin system inhibitors (RASi), sodium glucose co-transporter 2 inhibitors (SGLT2i), non-steroidal mineralocorticoids receptor antagonists (MRAs), and glucagon like peptide 1 receptor agonists (GLP1-RAs).

## 1. Introduction

Diabetes mellitus (DM) and chronic kidney disease (CKD) pose substantial challenges for older adult patients. In this review we aim to clarify emerging terminology, challenges in the diagnosis of CKD in older adult patients with diabetes, and a comprehensive approach to the management of such patients based on the current guidelines and emerging clinical trial data. Historically known as diabetic nephropathy (DN), CKD attributed to DM is often referred to as diabetic kidney disease (DKD) and is the major cause of CKD and end-stage kidney disease (ESKD) in those over 60 years old [1]. Furthermore, a third of new ESKD cases in those over 75 years old are attributed to DKD, almost systematically without a histologic diagnosis. For those individuals the more encompassing term is diabetes in CKD [2], a term that extends the classical definition of DKD [3] to reflect the complex and often multifactorial pathogenesis of CKD in individuals with diabetes. This form of CKD is increasingly not associated with proteinuria [4,5,6,7], and conversely, the presence of proteinuria or CKD in an older adult patient with DM is not always due to DN or DKD. Since none of the randomized controlled trials that led to therapeutic breakthroughs in this condition required a histological diagnosis for enrollment, the more general term more accurately reflects the population of patients studied in the trials and thus the evidence. In this review, we will use these terms interchangeably, noting that the more general term that aligns with the clinical trial enrollment criteria may be used with increasing frequency in the future.

Providing care in older individuals with DM and CKD is challenging because of the substantial comorbidity burden and the co-existence of physical and mental conditions such as dementia [8]. Multidisciplinary medical management may be required due to these concomitant conditions, and the application of emerging evidence-based therapies should neither be reflexively applied nor arbitrarily withheld.

While guidelines have incorporated fixed, age-independent referral criteria to nephrology, older patients may benefit less from nephrology referral than younger individuals [9], while the likelihood of regression, i.e., a spontaneous improvement in kidney filtration, often exceeds the likelihood of progression [10]. This creates challenges in translating interventions that are often tested in populations with some degree of proteinuria to those patients with (near-) normoalbuminuria. However, recent advancements in DKD therapy are applicable to both older and younger patients and confer broad cardiovascular benefits (e.g., on cardiovascular disease or heart failure) that may be particularly applicable to older patients with multiple comorbidities. Furthermore, these newer therapies are extremely easy to apply and can be used in a pillars of therapy approach by either primary care providers or specialists, offering unique opportunities to improve the outlook of older patients with DKD. In this review, we will focus predominantly on agents that exert cardiovascular and kidney benefits in individuals with CKD and DM, i.e., SGLT2i, GLP1-RAs, and MRAs, omitting agents such as the dipeptidyl peptidase 4 inhibitors whose cardiovascular and kidney safety has been demonstrated in randomized clinical trials but that are currently not indicated to reduce the high cardiovascular and kidney risk in patients with DM and CKD. 

## 2. Definition, Epidemiology, and Burden of Diabetes in Older Adult Patients with CKD

CKD is defined as either a persistent estimated GFR < 60 mL/min/1.73 m^2^ and/or evidence of kidney damage (most commonly albuminuria). Based on this definition and using NHANES data, an estimated 14% of the adult population has CKD; however, the prevalence of persistently decreased eGFR was ~29% in those 65 years old and older [11,12]. The epidemiology of CKD seems to vary with age, in that CKD determined by low eGFR alone (<60) is more common in older adults than in younger populations. Furthermore, the association of low eGFR with mortality and ESKD in older adults, while still present, is less pronounced in older individuals [13]. 

ESKD among patients with type II DM [4] is considered a problem of pandemic dimensions [14], because of the increasing rates of DM and obesity, which will further increase the prevalence of CKD and eventually the need for dialysis and kidney transplant. Sequential analyses of the National Health and Nutrition Examination Survey (NHANES analysis; 2003–2004 vs. 2013–2014) showed that the estimated frequency of DM increased by 9 million, affecting 30.2 (13%) million US adults [4]. The 2020 updated National Diabetes Statistics that examined data from 2013 to 2018 suggests very little improvement in this figure [15]. According to the same report the prevalences of diagnosed and total DM among people over the age of 65 in the United States were 21.4% and 26.8%, respectively. These worrisome trends are not US specific; the International Diabetes Federation Atlas [16] projects an increasing trend of DM up to 2045. The NHANES analyses also showed very little change in the prevalence of CKD in patients with diabetes as defined by a persistent urine albumin-to-creatinine ratio of ≥30 mg/g or a persistent eGFR of <60 mL/min per 1.73 m^2^, from 28.4% (1988 to 1994) to 26.2% (2009 to 2014) [11]. Data from the CURE-CKD registry, which collects electronic health data from two health care systems in the western US, provide a slight contrast to these trends. Based on this registry, the overall prevalence of CKD has trended down slightly among those with DM overall and noticeably in the oldest population when comparing 2015-2016, 2017–2018, and 2019–2020 [17].

The higher rates of DM pose a significant threat to the falling incidence rates of ESKD observed in recent years [18]. While improvements in the overall standards of care could reduce the risk of a single individual to experience worsening of their kidney function to the point they need dialysis or kidney transplant, the overall number of patients needing renal replacement therapy may increase if more patients develop DM. Analyses in NHANES [4] and in non-US cohorts [19] suggest that this epidemiological trend may be particularly relevant for older patients since those older than 65 years old are 35% more likely to manifest albuminuria than those younger than 65 (32.3% vs. 23.9%, NHANES estimate). Prior to the COVID19 pandemic, these trends were expected to lead to a rising prevalence in ESKD from 2015 to 2030 [20]. African Americans, Hispanic Americans, and American Indians [4], may exhibit even higher rates of CKD and DM in CKD [21], and these disparities carry over to self-management of DM among older Medicare beneficiaries [22]. 

## 3. Pathophysiology of CKD in DM and DKD among Older Adults

The natural aging process in the kidney shares histopathologic findings, e.g., sclerosis, mesangial matrix expansion, tubular and glomerular basement thickening, and interstitial atrophy and fibrosis [23], with those observed in DKD (Table 1), suggesting shared but not identical pathophysiology. Representative images from such lesions may be found in the American Journal of Kidney Disease Atlas of Renal Pathology [24] or from the PathologyOutlines.Com [25] atlas.

Hyperfiltration is thought to precede the functional loss of nephrons in DN; however, in “normal aging” this loss [29] is not accompanied by a rise in single-nephron GFR [30,31]. Genetic studies have shown that loci linked to albuminuria in aging mice and diabetic people are partially overlapping [32]. Accelerated senescence in tubular cells (primarily) and podocytes (secondarily) has been shown in kidney biopsies from patients with DN and type II DM [33]. Furthermore, these senescent cells secrete a variety of mediators (e.g., proinflammatory cytokines and complement component and pro-fibrotic factors) that drive glomerulosclerosis, podocyte hypertrophy, mesangial changes, and tubulointerstitial fibrosis observed in DKD [34]. Advanced glycosylation end products (AGEs) and the RAGE (cell surface receptor of AGEs) accumulate in both diabetic and senescent kidneys [35], where they promote oxidation and inflammation [36,37], thus increasing the likelihood of age- and diabetic-related CKD [38]. A recent review of the molecular pathways underlying the progression of DKD in older adults [39] revealed a complex interplay between oxidative stress, inflammation, and hyperglycemia.

The final common pathway linking inflammation and tissue fibrosis in CKD may be mediated through the aberrant activation of the mineralocorticoid receptor, an observation that is more than eighty years old [40]. Mineralocorticoid receptor activation links tissue injury, oxidative stress, inflammation, arterial hypertension, and end organ damage of the heart, blood vessels, and the kidneys [41,42,43]. 

While kidney biopsies are typically obtained not to diagnose DKD or DN but to exclude other non-diabetic forms of kidney injury, if diabetic lesions are obtained, they should be staged in both the glomerular and the vascular/tubulointerstitial compartments [26]. Due to the shared histology and pathophysiology the cause of the lesions observed in biopsies of patients with DM cannot be unequivocally assigned to the latter. For example, renal artery stenosis and secondary kidney ischemia [44] may lead to intrarenal arterial hyalinosis [45]. While efferent arteriolar hyalinosis is typically associated with a diabetic kidney lesion, afferent arteriolar hyalinosis may be observed in hypertensive nephropathy. Normoalbuminuric kidney disease in the setting of DM is becoming a more common clinical phenotype; it may be associated with non-glomerular lesions [46,47] or the successful use of antiproteinuric therapies such as inhibitors of the renin angiotensin system [4,48,49,50].

## 4. Diagnosis of CKD in Older Patients with DM

The diagnostic criteria for CKD do not vary by change, i.e., either a depressed eGFR or an elevated index of protein excretion in the urine (usually measured as the ratio of urinary albumin to urinary creatinine on a spot urine specimen) as a marker of kidney damage should be demonstrated for the diagnosis and staging of CKD. In caring for individual patients, the potential for an age-related loss of kidney function should also be considered as a potential cause of a reduced eGFR value in the absence of albuminuria. One cohort study was conducted in Canada with a considerable representation of individuals ≥65 years old and compared the implications of age-adapted vs. fixed eGFR on the 5-year risk of renal failure and death. They identified a similar difference in the 5-year absolute risk of kidney failure (0.12%) among individuals 65 years and older who had an eGFR of 45–60 mL/min/1.73 at baseline and no detectable proteinuria compared with non-CKD patients [51]. Proposals for an age-adapted definition of CKD [52], by adopting a threshold of <45 mL/min/1.73 m^2^ instead of <60 mL/min/1.73 m^2^, have been proposed (Figure 1), but at the present time, the guidelines endorse only the fixed age-independent threshold of 60 mL/min/1.73 m^2^ (Figure 1).

Albuminuria is never a manifestation of “normal aging”, and its presence signifies an elevated risk for the progression of CKD, endothelial dysfunction, need for dialysis, and cardiovascular morbidity and mortality [54,55,56]. For the older individual with DM, who is at risk for other forms of kidney disease (e.g., vasculitis), the initial diagnostic step is to exclude a non-diabetic kidney lesion. In particular, if the diagnosis of CKD predates the diagnosis of DM or occurs within a short period of time (5–10 years), then the risk for another kidney disorder is particularly high [57]. 

The initial workup should not differ for older and younger patients and includes a urinalysis, urine albumin to creatinine ratio (UACR), an eGFR, a complete blood count, and a basic metabolic profile that incorporates measurements of sodium, potassium, bicarbonate, calcium, and phosphorus [2,3]. Serological tests as per the guidance of the US National Institute for Diabetes, Digestive and Kidney Diseases include tests for chronic hepatitis B and C, antinuclear antibodies, rheumatoid factor, complement levels (C3/C4), serum and urine protein electrophoresis, and a free light chain assay. A kidney ultrasound is also part of the workup and can be used to diagnose bona fide urinary outflow obstruction (e.g., hydronephrosis) or subtler forms of bladder dysfunction (e.g., an elevated postvoid residual urinary volume in the bladder). If a patient with DM has typical and advanced retinopathy [58,59,60,61], albuminuria, and a negative serologic evaluation, most clinicians would not proceed to obtain a kidney biopsy. In the older patient, vascular disease (e.g., due to atherosclerosis, hypertension, and RAS) [44,62] may also be present, and such conditions may be used to diagnose the patient with cardiovascular disease and target them for high-intensity therapy to reduce cardiovascular risk. At the time of this writing, a precise histological diagnosis of DKD is not required to initiate therapies such as inhibitors of the renin angiotensin system or sodium glucose co-transporter 2 inhibitors, whose spectrum of indications includes both diabetic and non-diabetic kidney lesions. However, a missed glomerular diagnosis does not allow the initiation of specific therapy that may preserve kidney function or prevent damage to other organs (e.g., due to vasculitis). Since the histology cannot be predicted from clinical criteria [63] and a definitively higher risk of bleeding is not seen in older adults [64,65,66], it may be reasonable to apply the same “atypical feature” [67,68,69,70] criteria for pursuing a kidney biopsy as in the young (Table 2). Of note, there is no specific eGFR cutoff for a kidney biopsy; rather, the procedure is pursued when the patient is considered at risk for a non-diabetic kidney lesion (e.g., an active urinary sediment with hematuria or casts) or when there is accelerated kidney functional decline, which merits ruling out other diagnoses (such as rapidly progressing glomerulonephritis or vasculitis). 

## 5. Treatment Considerations

The general approach to treating CKD in DM in older adults is not different from the one applied to younger individuals, although the elements should be highly individualized to account for other medical problems and comorbidities with advancing age. When approaching any patient with CKD, the overarching aim is to control the risks of both cardiovascular disease and kidney disease progression as most patients are more likely to experience a cardiovascular event than to need dialysis [71,72]. The components of this approach include lifestyle changes (smoking cessation and a healthy lifestyle that includes exercise preferably longer than 150 min weekly), a reduction in sodium intake (to less than 2 g every day) and avoiding extreme protein intakes (e.g., 0.8 gm/kg/d is a reasonable goal), and active pharmaceutical interventions to control blood pressure, glycemia, atherosclerotic cardiovascular risk, and specific antiproteinuric and antifibrotic therapies [2,3,72]. 

### 5.1. Non-Pharmaceutical Interventions and Goals of Therapy

#### 5.1.1. Exercise

An individualized, planned, and supervised combination of aerobic and resistance training is considered the most effective way to control glycemia [73]. Contraindications to exercise stem from specific comorbidities (e.g., proliferative retinopathy, aneurysms, or severe autonomic insufficiency with propensity for hypoglycemia) and are often temporary in nature (e.g., during an acute exacerbation of ischemia, heart failure, or hypertensive event and periods of poor glycemic control with propensity to hypoglycemia) [74]. Tailoring the prescription of exercise program to the functional capacity limitations of the individual can be achieved through the Vivifrail multicomponent exercise program [75].

#### 5.1.2. Dietary Considerations

Sodium restriction to less than 2 g a day improves hypertension and the efficacy of antihypertensive medications. When the dietary intake of fresh vegetables is poor, increasing levels of sodium intake have been associated with an increased incidence of diabetic retinal disease [76]. A DASH diet can improve hypertension control but can lead to hyperkalemic episodes in individuals with hyporeninemic hypoaldosteronism. Current American Diabetes Association (ADA) guidelines suggest limiting protein intake to 0.8–1.0 g/kg/day in those with DM and CKD. Nevertheless, studies on dietary protein restriction have failed to show a clear benefit in reducing the progression of DKD [62,77,78]. One should be aware that this target conflicts with that in the guideline for older adults that recommend a target protein intake of 1–1.2 g/kg to prevent malnutrition and sarcopenia in old age [79]. The benefit (progression of CKD) to risk (developing protein-energy wasting) should be carefully balanced when prescribing low-protein diets in older adults with advanced CKD. This balancing act should consider the presence of pre-existing malnutrition, the rate of progression of CKD, the CKD stage, and the presence of comorbidities/anticipated life expectancy, which would shift the focus toward quality rather than quantity of life and dialysis avoidance. An approach that attempts to reconcile guidelines and prioritize goals distinguishes between “nutritional-geriatric” priorities (age is the dominant factor) vs. “renal” priority (when the patient’s goal would be to avoid dialysis at any cost), emphasizing that these priorities may shift during periods of critical illness [80]. Implementing nutritional therapies should be performed in a stepwise manner: first, a nutritional assessment with a validated tool should take place, and if the patient screens positive for protein-energy malnutrition, a formal assessment with the Subjective Global Assessment (SGA) should be performed; second, an ongoing evaluation of muscle mass and function during the implementation of dietary restrictions to detect early development of sarcopenia and protein-energy malnutrition that would limit the continuation of such diets should follow [80]. 

#### 5.1.3. Blood Pressure, Lipid, and Glycemia Control in Older Adults with CKD in DM

The ADA standards of care in the DM [81] framework consider blood pressure, glycemic control, and lipids together, and this integrative, comprehensive approach provides a solid base to approach the older individual with CKD in DM (Table 3). This framework acknowledges that tight glycemic control comes with higher risks [82,83,84,85,86] in such individuals due to a non-robust physiologic response to hypoglycemia and greater hypoglycemia unawareness. That framework progressively de-escalates the aggressiveness of goal-directed therapy as the number of pre-existing conditions (those severe enough to require medication or lifestyle management) advance to end-stage chronic illness. The blood pressure targets that the ADA proposes differ from those in the KDIGO guidelines [87], which are heavily influenced by the SPRINT trial [88]. However, even the KDIGO guidelines point out that the benefits of tight blood pressure control are less certain in those with DM and advanced (stages 4 and 5) CKD and the very old (individuals older than 90). Hence, for older patients with DM and CKD who find themselves at the intersection of these groups, shared decision making and individualized goal setting that minimizes the side effects of therapy and their effect on quality of life should take precedence over a “one-size-fits-all” blood pressure target.

When statins are used for primary and secondary prevention, a benefit can be unequivocally expected for those individuals whose life expectancy exceeds the time frames (2–6 years) of the clinical trials [89]. Equivalently, one may use the average time to benefit for a therapy, which for statins was 2.5 years [90], and treat individuals whose expected survival exceeds this time frame. Many advanced age individuals with CKD stage 3a–5 would benefit according to this criterion, and this is the reason the KDIGO clinical practice guidelines [91] recommend the use of a statin or a statin/ezetimibe in patients older than 50 years old. These recommendations are largely based on the SHARP trial [92] that randomized participants with CKD (mean eGFR of 27 mL/min/1.73 m^2^, *N* = 9270) to receive simvastatin 20 mg plus ezetimibe 10 mg daily or a placebo. Statin plus ezetimibe therapy reduced the primary outcome of major atherosclerotic event (coronary death, myocardial infarction, need for revascularization, or non-hemorrhagic stroke) by 17% (95% CI: 0.06–0.26) but without delaying dialysis. 

### 5.2. Pharmaceutical Interventions to Reduce Cardiorenal Risk in Older Patients with CKD in DM

At the present time, multiple agents have established a track record in reducing the adverse cardiovascular and renal effects of chronic kidney disease in DM/DKD. These agents target various structural elements in the kidney (Figure 2) while also having systemic sites of actions (e.g., inhibitors of the renin angiotensin system and mineralocorticoid antagonists also target the heart and the blood vessels, while the actions of the glucagon like peptides appear to be multisystemic and likely include both kidney and vascular targets).

#### 5.2.1. Inhibitors of the Renin Angiotensin System

Inhibitors of the renin angiotensin system (RASi) were established in the treatment of DKD by the pivotal trials IDNT [93] and RENAAL [94]. IDNT and RENAAL established that the benefits of RASi are maximized when the dose is maximized and identified residual albuminuria as a marker of increased cardiovascular and kidney disease risk [95,96] and a criterion for adding additional agents and/or enrolling patients in trials of investigational agents to retard the progression of kidney disease. In a recent analysis these agents were underutilized: 17% of patients with DM initiated these agents [97] within 12 months of diagnosis of CKD [98]; eventually, these agents are only used in ~60% of eligible patients but without any racial disparities in utilization [99]. The British Clinical Diabetologists and the UK Renal Association guidelines about the management of ACEi and ARBs in patients with DM and CKD [100], which do not explicitly apply to older individuals, are summarized below: Kidney function and potassium levels should be checked within 7 to 10 days after initiation.Up to 30% of eGFR decline may be tolerated.Drops in kidney function of more than 30% should prompt investigation for RAS, sepsis, volume depletion, or concomitant medications, e.g., NSAIDs.If an alternative explanation for a marked decline in renal function cannot be inferred, the dose of the RASi may be reduced.Potassium binders (Patiromer and sodium zirconium cyclosilicate) may be used to reduce the serum potassium if it rises over 5 mEq/L and allow the RASi to be continued.Combination therapy with ACEi, direct renin inhibitor, and ARBs should not be used, since multiple clinical trials have shown greater risks of hypotension, hyperkalemia, and acute renal injury with these combinations [101].In advanced (stage 4 and 5) CKD, discontinuation [102] of the RASi was associated with a lower risk for hyperkalemia (HR, 0.65; 95% CI, 0.54–0.79) but a higher risk of death (HR, 1.39, 95% CI 1.20–1.60) and a higher risk of progression to ESKD (HR 1.19, 95% CI: 0.86–1.65). The STOP-ACEi [103,104] RCT examined the benefits vs. harm of stopping the RASi in patients with advanced CKD (eGFR was ~18 mL/min/1.73 m^2^ at baseline). There was no difference in the eGFR (primary outcome) at 3 years between participants older than 65 years (−0.32, 95% CI −2.72–2.09 mL/min/1.73 m^2^) and those younger than 65 years (−0.32, 95%CI −2.92–2.28 mL/min/1.73 m^2^). ESKD occurred in 128 patients (62%) among those who discontinued the RASi and in 115 patients (56%) who continued them (HR, 1.28; 95% CI, 0.99 to 1.65). There were similar numbers of cardiovascular events (108 vs. 88) and deaths (20 vs. 22) in the two arms.

As noted above, RASi offer significant benefits to patients with DM and CKD; however, caution needs to be exercised with declining renal function particularly to avoid severe hyperkalemia. Overall, RASi remain one of the primary interventions in diabetic nephropathy to prevent the progression of CKD. An approximate 5-year risk reduction for ESKD of 30% with use of RASi is a useful rule of thumb, although specific benefits in patient subsets may vary [105].

#### 5.2.2. Sodium Glucose Co-Transporter 2 Inhibitors

SGLT2is are small molecules that act on the luminal side in the proximal tubule of the kidney and inhibit the SGLT2 transporter. In meta-analyses of treatment naïve patients and patients treated with metformin, SGLT2is reduced HbA1c by −0.81 to −1.02% and by −0.57 to −0.63%, respectively [106]. The glucosuric effect of this category of antidiabetic agents depends on the total eGFR, and they become less efficacious in reducing HbA1c when eGFR drops below 60 mL/min/1.73 m [107,108]. Nevertheless, according to Brenner’s hypothesis, hyperfiltering nephrons exist at all levels of kidney dysfunction, and SGLT2is continue to alleviate kidney hyperfiltration in diabetic patients with low GFRs. 

The cardiovascular safety trials for empagliflozin (EMPA-REG OUTCOME) [109,110], canagliflozin (integrated CANVAS program consisting of two clinical trials, CANVAS and CANVAR-R) [111,112,113], dapagliflozin (DECLARE-TIMI-58), and ertugliflozin (VERTIS-CV) [107,111] hinted at the combined cardiorenal benefit of SGLT2i. In these, SGLT2i reduced major adverse cardiovascular events (MACEs: cardiovascular death, non-fatal myocardial infarction, or stroke) in the case of the empagliflozin, and canagliflozin trials were non-inferior in the dapagliflozin and ertugliflozin trials. All drugs reduced heart failure hospitalizations in these trials. When the composite kidney-specific outcome of progression to dialysis dependency/need for kidney transplantation and declines in eGFR was harmonized across the four trials [107], SGLT2is were associated with renal benefits. Heart-failure-specific trials have included patients with reduced (dapagliflozin, DAPA-HF [114], and empaglifozin EMPEROR-REDUCED [115]) and preserved (dapagliflozin, DELIVER [116], and empagliflozin EMPEROR-PRESERVED [117]) ejection fractions. Trials with primary kidney-specific outcomes include the CREDENCE trial (canagliflozin) [118] in DKD, DAPA-CKD (dapagliflozin) [119], and EMPA-KIDNEY (empagliflozin) [120]. The latter two trials included patients with both diabetic and non-diabetic forms of CKD. All SGLT2i kidney outcome trials included patients who manifested residual, moderate to severe albuminuria while on the standard of therapy for kidney disease, i.e., a maximum tolerated dose of an ACEi or an ARB. Across all trials, the SGLT2i were associated with biphasic effects on the eGFR, with an acute dip of between 2 and 5 mL/min/1.73 m^2^ during the first month [121,122,123] followed by stabilization thereafter, while the participants in the placebo group experienced a faster decline in kidney function.

A previous random effect meta-analysis that modeled heterogeneity in these trials [124] by one of the authors has demonstrated that the beneficial effects of SGLT2i are a class effect. Other meta-analyses have shown that the benefits of these drugs do not vary by participant age [125,126]. In Figure 3 we summarize the age by subgroup results in the cardiovascular, heart failure, and kidney outcomes in the SGLT2i trials to date. Taken as a class, the interactions with age are not statistically significant (*p* = 0.294, Wald *p*-value), i.e., the benefits of the SGLT2i do not vary by age. Furthermore, there was no evidence of heterogeneity by drug type (*p* = 0.62, ANOVA test comparing a model adjusting for drug, outcome, age group vs. model adjusting for outcome, age group). SGLT2is are in general safe drugs, but reported side effects such as diabetic ketoacidosis and lower limb amputations appear to be barriers in prescribing them. A meta-analysis [127] has quantified the risks and benefits of SGLT2i in patients with and without DM: the number needed to treat to prevent one death (120) or one kidney disease progression event (48) dominated the number needed to harm for the development of one lower limb amputation (309) or diabetic ketoacidosis event (636). For most patients, SGLT2i would accrue a 3- to 10-fold larger benefit than risk depending on the specific pair of outcomes considered [127]. Of note, acute kidney injury (AKI) is reduced by 23% (RR 0.77, 95% CI 0.70–0.84) under an SGLT2i. A framework for managing the risks of the SGLT2i was put forward in a roundtable discussion involving representatives from cardiology, endocrinology, and nephrology (Table 4). 

#### 5.2.3. Mineralocorticoid Antagonists

The use of mineralocorticoid receptor antagonists (MRAs) in CKD with DM is predicated on their anti-inflammatory and anti-fibrotic effects on the heart, blood vessels, and kidneys. The benefits of steroidal MRAs (e.g., eplerenone and spironolactone) were summarized by the Cochrane group [129] and included reductions in systolic blood pressure by ~5 mmHg (95% CI −8.22 to −1.75 mmHg) and protein excretion by ~0.5 g per day (95% CI −0.2 to −0.82 g/day) but had uncertain effects on kidney failure, cardiovascular, and total mortality. In this meta-analysis of mostly spironolactone studies, the risks for gynecomastia and hyperkalemia were increased with the use of spironolactone. 

Non-steroidal MRAs such as finerenone, esaxerenone, and apararenone offer a balanced antagonism in the kidney and the heart, thus reducing the risk of hyperkalemia [43,130]. Phase 2 clinical trials with esaxerenone [131] and apararenone [132] in DKD show that these agents may reduce proteinuria by 40–60% when added on RASi. At the time of this writing the only commercially available non-steroidal MRA for CKD in DM is finerenone. The approval of finerenone was based on two large randomized controlled trials, FIDELIO-DKD [133] and FIGARO-DKD [134], and a pre-specified patient-level meta-analysis of these two trials (FIDELITY) [135], which provided the data for the effects of this drug on cardiovascular and kidney-specific outcomes. Both these studies followed a similar design, i.e., they enrolled patients with CKD in type II DM who had some degree of residual albuminuria despite being on a maximum tolerated dose of RASi. FIGARO-DKD recruited patients with better preserved kidney function (UACR > 300 mg/g with eGFR > 60 mL/min/1.73 m^2^ or UACR in 30–300 mg/g and eGFR in 25–90 mL/min/1.73 m^2^), while FIDELIO-DKD recruited patients with more advanced CKD (UACR > 300 mg/g and eGFR 25–75 mL/min/1.73 m^2^ or UACR in 30–300 mg/g and eGFR 25–60 mL/min/1.73 m^2^). The primary outcome of FIDELIO-DKD was a composite of kidney failure (need of dialysis and transplant) and a sustained decrease in the eGFR by 40% relative to the baseline and death from renal causes. The primary outcome for FIGARO-DKD was a composite of cardiovascular death, non-fatal myocardial infarction and stroke, and hospitalization for heart failure (MACE/HHF). The primary outcome of FIGARO-DKD was a secondary outcome of FIDELIO-DKD and vice versa, enabling the joint examination of the effects of finerenone on the cardiorenal risk in patients with DKD in FIDELITY. Finerenone was equally effective in younger (<65 years old) and older (≥65 years old) patients (Table 5). None of the differences were statistically significant at the 0.05 level. 

#### 5.2.4. GLP1 and Dual GLP1/GIP1 Receptor Agonists

GLP1 and the dual receptor agonists of the GLP1/GIP receptors are a class of antiglycemic agents with broad cardiometabolic effects and emerging kidney and cardiovascular benefits. These drugs activate the receptors of the endogenous incretins, glucagon-like peptide 1 and glucose-dependent insulinotropic polypeptide (GIP). Similar to SGLT2is, GLP1-RAs were initially introduced to reduce glycemia with a minimal risk for hypoglycemia, while also reducing weight. Specific GLP1-RAs (liraglutide and semaglutide) have been approved as anti-obesity medications even in patients without DM. Dual agonists are associated with more pronounced weight loss [137] and an enhanced antiglycemic effect relative to insulin or pure GLP1-RAs in the SURPASS clinical trial [138,139,140,141]. Triple antagonists (GLP1/GIP1 and glucagon receptor antagonists) are also entering the therapeutic arena [142], but the dedicated cardiovascular and kidney benefit trials have not been reported yet, and the agents are not available yet for clinical use. Certain GLP1 classes (dulaglutide, liraglutide, and semaglutide) have been shown to reduce cardiovascular disease and thus are indicated in the ADA standards of care for DM [143] for the management of patients with atherosclerotic cardiovascular disease (ASCVD) or with high-risk indicators of ASCVD. In a recent meta-analysis [126], GLP1-RAs in adults older than 65 years old were associated with a 15.3% (OR 0.85, 95% CI 0.79 to 0.91) reduction in MACE events, similar to the 16% (OR 0.84, 95% CI 0.70 to 1.01) benefit seen in younger adults. Hence, GLP1-RAs are equally beneficial in older and younger adults with type II DM for the management of their cardiovascular disease. To date, the clinical benefits of GLP1-RAs and GLP1/GIP RA on kidney outcomes have been limited to surrogate markers of kidney function loss (eGFR slope) and markers of kidney damage (UACR) often examined as explorations of kidney-specific outcomes [144,145,146,147] in their cardiovascular safety and primary efficacy trials. A recent meta-analysis [148] examined the effects of GLP1-RAs on cardiovascular (MACE) and two kidney outcomes: a kidney composite consisting of development of macroalbuminuria, doubling of serum creatinine or at least a 40% decline in eGFR, kidney replacement therapy, or death due to kidney disease, and worsening of kidney function, defined as either doubling of serum creatinine or at least a 40% decline in eGFR. GLP1-RAs reduced MACE by 14% (HR 0.86, 95% CI 0.80–0.93, *p* < 0.0001), with no evidence of interaction by age (*p* = 0.78 comparing effects in individuals younger than 65 vs. those older than 65). Age-stratified kidney outcomes were unavailable in this paper; however, overall, GLP1-RAs were associated with a favorable effect on the composite kidney outcome (21%reduction (HR 0.79 [95% CI 0.73–0.87]; *p* < 0.0001)), with a favorable trend for the worsening kidney function outcome (14% reduction (HR 0.86 [95% CI 0.72–1.02])). In REWIND [144], one of the few GLP1-RAs trials to report a kidney-specific outcome by age, individuals older than 66 years had an HR of 0.79 (95% CI 0.69–0.90) that was not statistically different (p-value for the interaction 0.17) from that of individuals younger than 66 (HR: 0.90, 95% CI: 0.79–1.02). The kidney outcome in REWIND was a composite of the development of incident macroalbuminuria (300 mg/gm of creatinine) and a sustained (>30% in two consecutive measurements) decrease in eGFR from the baseline or new kidney replacement therapy, i.e., a broad outcome that considered both the development of worsening kidney damage (UACR elevation) and function (decreased eGFR/need for dialysis), as in the SGLT2i and MRA trials. In SURPASS-4 [145], the placebo-corrected difference in the eGFR slope did not differ in older (≥65 years old) and younger individuals: 2.4 (1.5–3.3) vs. 2.1 (95% CI 1.2–2.9) ml/min/1.73 m^2^/year, p for interaction = 0.67. Additionally, the least squares change from the baseline over the placebo was −38.5% (95% CI: −43.6% to −26.2%) vs. −28.5% (95% CI: −36.4% to −19.7%), p for interaction = 0.80. Hence, similarly to SGLT2i and non-steroidal MRA, the beneficial effects of GLP1 and GLP1/GIP receptor agonists are observed across the adult age span. Tirzepatide (GLP1/GIPRA) and Retartrutide (GLP1/GIP/GlucagonRA) are examined in ongoing trials to assess their impacts on renal function in overweight patients with and without DM (NCT05936151 and NCT05536804). 

## 6. Conclusions

CKD in older patients with DM is associated with both cardiovascular risks and kidney progression risks. The pathophysiology is complex, and the spectrum of CKD is not limited to typical DN but may also include additional vascular insults superimposed on a senescence molecular phenotype. Care of the older patient with CKD in DM requires a multidisciplinary, holistic approach that considers cardiovascular risk, comorbidities, and life expectancy in addition to the risk of kidney disease progression. The effects of emerging standard of care pharmaceutical interventions (e.g., SLGT2 inhibitors, MRAs, and GLP1-RAs /dual GLP1/GIP RA antagonists) to reduce the risk of cardiovascular and kidney risk are observed in both younger and older individuals. Hence, there should be no age discrimination in prescribing these newer agents for older individuals as the trials to date suggest that older individuals will derive benefit from such therapies. Finally, meta-analyses [149,150] and actuarial re-analyses of outcomes trial data [151] suggest that the combination of SGLT2i/ GLP1-RAs and SGLT2i/MRA/ GLP1-RAs will be associated with improved glucometabolic control and cardiovascular/dialysis-free and overall survival. Considering the high risk for cardiovascular death and the detrimental effects of dialysis upon the quality and quantity of life among older adults, strong consideration should be given to combination therapies with these classes of agents.

## Figures and Tables

**Figure 1 jcm-13-00348-f001:**
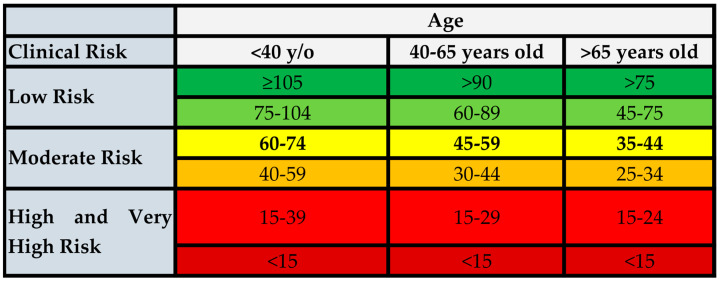
Age-adapted definition of CKD based on eGFR risk categories according to patient’s age. Risk categories were defined in re-analyses of the CKD Prognosis Consortium data using mortality as the outcome. These analyses suggest the need for a higher eGFR threshold for the diagnosis of CKD in younger individuals (<75 instead of <60 mL/min/1.73 m^2^) and a lower eGFR value in older individuals [52,53]. Figure 1 summarizes these proposals using the color-coding system in the current KDOQI guidelines. In this color-coding scheme, the lowest category of risk is indicated by the green color, and the highest risk with increasing shades of red.

**Figure 2 jcm-13-00348-f002:**
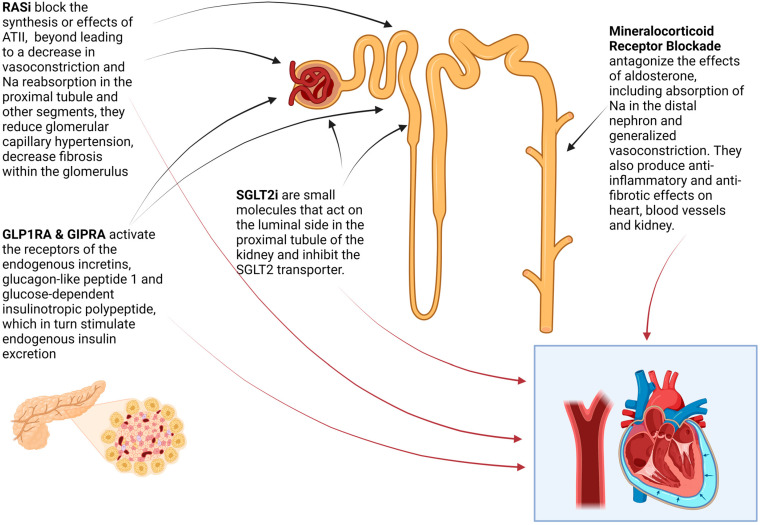
Mechanisms of action of key pharmacologic interventions for CKD in DM. Figure created by Biorender. RASi: Renin Angiotensin System inhibitors (Section 5.2.1) Examples of commercially available agents and brand names in these classes include GLP1-RAs (dulaglutide (Trulicity), liraglutide (Victoza/Saxenda), and semaglutide (Ozempic/Rybelsus/Wegony)), GLP1-RAs /GIPRA (tirzepatide (Zepbound/Mounjaro), discussed at Section 5.2.4, mineralocorticoid receptor antagonists (finerenone (Kerendia)) discussed at Section 5.2.3, SGLT2i (bexagliflozin (Brenzavvy)), canagliflozin (Invokana), dapagliflozin (Farxiga/Forgixa), empagliflozin (Jardiance), and ertugliflozin (Steglatro) discussed at Section 5.2.2.

**Figure 3 jcm-13-00348-f003:**
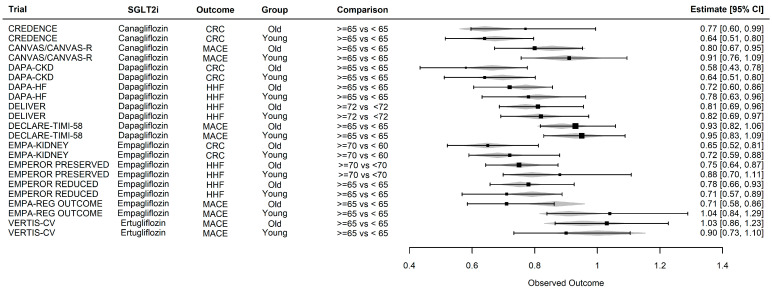
SGLT2i and clinical outcomes in older vs. younger individuals (hazard ratio and 95% confidence intervals). CRC: cardiorenal composite (CREDENCE: death from renal or cardiovascular causes, doubling of serum creatinine, or kidney failure defined as eGFR < 15 mL/min/1.73 m^2^, need for dialysis or transplant, DAPA-CKD: death from renal or cardiovascular causes, decline of >50% of the eGFR from baseline and kidney failure, defined as need for dialysis, transplant, or sustained eGFR to less than 15 mL/min/1.73 m^2^, EMPA-KIDNEY: death from cardiovascular cases or progression of kidney disease defined as ESKD, sustained decrease in eGFR <10 mL/min/1.73 m^2^, decrease of eGFR >40% from baseline, death from kidney renal causes), HHF: hospitalization for heart failure, MACE: major adverse cardiovascular events (composite of cardiovascular death, non-fatal myocardial infarction, or stroke). Diamonds show predictions for a random effects model adjusting for drug, outcome, and age subgroup, and whiskers and boxes show the observed hazard ratios and 95% confidence intervals in the source trial data.

**Table 1 jcm-13-00348-t001:** Lesions by histologic compartment in the 2010 Pathologic Classification of Diabetic Kidney Disease [26].

Glomerulus	Arterioles	Mesangium ^2^	Tubulo-Interstitium
Diffuse intra-capillary glomerulosclerosis	Subintimal hyaline deposits	Mesangial matrix expansion	Tubular atrophy/interstitial space expansion
Nodular (Kimmelstein Wilson)	Capillary walls, Bowman capsules (capsular drops) ^1^	Mesangiolysis	Tubular basement membrane thickening
Glomerulosclerosis		Mesangial cell proliferation	Interstitial fibrosis

^1^ Capsular drops may be observed in a small percentage (~5%) of patients without diabetes [26] but are considered in general specific for DKD [27]. ^2^ Mesangial lesions correlate with loss of eGFR albuminuria and hypertension [28].

**Table 2 jcm-13-00348-t002:** Features atypical of diabetic kidney disease: when to consider additional or alternative pathology and biopsy.

Features on Presentation	Features Developing on Presentation or Follow-Up
Absence of retinopathy	Rapid decline in eGFR (>5 mL/min/1.73 m^2^/year)
Albuminuria <5 years or >25 years after the diagnosis of type 1 diabetes	↓ eGFR by more than 30% after initiation of an inhibitor of the renin angiotensin system
Active urine sediment or serologies	Acute kidney injury (unexplained/sustained)
Hematuria/nephritic syndrome	Sudden/acute worsening of albuminuria (unexplained/sustained)

**Table 3 jcm-13-00348-t003:** Targets of anti-hypertensive, glycemic, and lipid therapy in the older patient with CKD in diabetes.

	Healthy	Complex	Very Complex
Patient characteristics Health status	Few coexisting chronic illnesses and intact cognitive and functional status	At least three coexisting chronic illnesses or 2+ instrumental ADL impairments or mild-to-moderate cognitive impairment	Long-term care facility resident or end-stage chronic illnesses or moderate-to-severe cognitive impairment or 2+ ADL impairments
Rationale	Longer remaining life expectancy	Intermediate remaining life expectancy, high treatment burden, hypoglycemia vulnerability, fall risk	Limited remaining life expectancy makes benefit uncertain
HbA1c	<7.0–7.5% (53–58 mmol/mol)	<8.0% (64 mmol/mol)	Do not rely on HbA1C; glucose control decisions should be based on avoiding hypoglycemia and symptomatic hyperglycemia.
Fasting/pre-prandial glucose	80–130 mg/dL (4.4–7.2 mmol/L)	90–150 mg/dL (5.0–8.3 mmol/L)	100–180 mg/dL (5.6–10.0 mmol/L)
Bedtime glucose	80–180 mg/dL (4.4–10.0 mmol/L)	100–180 mg/dL (5.6–10.0 mmol/L)	110–200 mg/dL (6.1–11.1 mmol/L)
Blood pressure	<140/90 mmHg	<140/90 mmHg	<150/90 mmHg
Lipid target	Statin unless contraindicated or not tolerated	Statin unless contraindicated or not tolerated	Consider likelihood of benefit with statin

Coexisting chronic illnesses: arthritis, cancer, heart failure, depression, emphysema, falls, hypertension, incontinence, stage 3 or worse CKD, myocardial infarction, and stroke. End-stage chronic illness, such as stage 3–4 heart failure or oxygen-dependent lung disease, dialysis-dependent ESKD, or uncontrolled metastatic malignancy. ADL: activities of daily living. Definitions of older adults vary somewhat in the literature, and a clear distinction between older adults ≥65 years of age as opposed to someone ≥90 years is often made. It may be prudent to view individuals ≥90 years as more complex, even if they are otherwise healthy.

**Table 4 jcm-13-00348-t004:** Adverse events associated with SGLT-2i and proposed preventative measures.

Adverse Events	At Risk	Preventive Measures
Genitourinary infections	Women, uncircumcised men	Adequate perineal hygiene Optimal diabetes care Antifungals Avoid SGLT-2is in patients with history of severe, recurrent infections
Diabetic ketoacidosis	Insulin deficiency, ketogenic diet, alcohol abuse, acute illness, surgery	Maintain insulin; ≤20% reduction in insulin dosage if necessary Discontinue SGLT-2i temporarily in acute illness or surgery Avoid SGLT-2is in patients with history of DKA Discontinue SGLT-2i if patient is not eating or has vomiting and/or diarrhea
Acute kidney injury	eGFR dip ≥30%, volume depletion	Reassess SGLT-2i regimen Frequently assess renal function, especially in patients with baseline eGFR <60 mL/min/1.73 m^2^ Discontinue SGLT-2i temporarily in acute illness
Volume depletion	eGFR <60 mL/min/1.73 m^2^, old age, concomitant diuretic, prior volume depletion, hypotension, SBP <110 mm Hg	Reduce diuretic or hypotension-inducing agent use Inform patients to maintain adequate oral hydration Discontinue SGLT-2i temporarily in AKI
Hypoglycemia	Concomitant insulin or SU, old age	Reduce insulin ≤20% or SU ≤50% if HbA1c <7.0%–8.0% Discontinue SU if HbA1c <8.0% in older patients Gradually reduce SU if HbA1c <8.0% in younger patients
Amputation	History of amputation, peripheral vascular disease, neuropathy, foot ulcers	Monitor at-risk patients for new pain, skin ulcerations, or infections Inform patients about proper foot care
Hyperkalemia	No concern	

The data in this table were adapted from Figure 3 from [128] under the Creative Commons Attribution (CC BY) license.

**Table 5 jcm-13-00348-t005:** Finerenone and clinical outcomes in older vs. younger individuals (hazard ratio and 95% confidence intervals).

Clinical Trial	Outcome	Effect in Younger Patients	Effect in Older Patients
FIGARO-DKD	MACE/HHF ^2^	0.90 0.74–1.10	0.85 0.72–1.00
FIGARO-DKD ^1^	CR ^3^	0.72 0.52–0.99	0.92 0.61-1.38
FIDELIO-DKD	CR	0.85 0.72–1.01	0.79 0.67–0.94

^1^ The subgroup analysis was presented in a follow-up publication [136] and used a sustained reduction of eGFR > 57%, rather than the 40% used in the primary analysis of the FIGARO-DKD study. ^2^ HHF: hospitalization for heart failure, MACE: major adverse cardiovascular events (composite of cardiovascular death, non-fatal myocardial infarction, or stroke). ^3^ CR: composite renal outcome.

## Data Availability

Data are contained within the article.

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
