# Peer review of "Chronic Kidney Disease in the Older Adult Patient with Diabetes"

_jcm, 2024, doi:10.3390/jcm13020348_

Round 1
Reviewer 1 Report
Comments and Suggestions for Authors
In the review titled "Chronic Kidney Disease in the Elderly Patient with Diabetes," the authors have summarized the current treatment options for CKD patients, both with and without diabetes mellitus (DM), and in different age groups. They discussed new therapies and optimal management for CKD patients, with a focus on elderly individuals. To enhance the review, consider the following suggestions:
1. Introduction Clarity: The introduction should be more straightforward. In lines 32-37, specify the type of CKD being discussed, its differences in patients with diabetes (DKD) vs. normo-albuminuria, CKD in elderly vs. young patients, current therapies, their limitations, and the objectives of the review.
2. This review would benefit from the inclusion of summary figures or flowcharts. For instance, Table 1 could be improved by incorporating histopathologic images or a model illustrating the pathologic classifications of DKD and highlighting the distinctions from non-DKD and aging, supported by the latest references.
3. Staging Table: consider creating a table that stages CKD in different age groups to assist readers.
4. Authors should consider discussing gender differences in CKD and their care management.
5. Treatment considerations: Authors should consider making a table – summarizing general approaches for treating CKD in elderly with or without DM, young people, with other comorbidities and include proper references. This would help separate clarity what authors are proposing later in this review as individual care managements
6. Pharmacological Interventions: Include a figure or model detailing various pharmacological interventions, their sites of action, and mechanisms.
7. DPP-4 Inhibitors: Discuss DPP-4 inhibitors, their mode of action, and relevant clinical trials.
8. GLP1R/GIPR/Glucagon Triple Agonist: Consider updating the review with information on the ongoing study involving Retatrutide (LY3437943) or its potential effects on CKD. Additionally, discuss the potential benefits of a combination of GLP1R/GIPR agonists with SGLT2 inhibitors (e.g., Tirzepatide). Please review and incorporate following references,
Clinical trials : NCT05936151; NCT05536804;
Irene Caruso et al. eClinicalMedicine. 2023 Oct; 64: 102181.
Ania M. Jastreboff et al. N Engl J Med 2022; 387:205-216
Shweta Urva et al. The Lancet, Volume 400, Issue 10366,2022, Pages 1869-1881,
9. “… and propose a framework for the management of DKD…” this proposal is missing at the end.
Minor Corrections:
- Rewrite the sentence at line 63-64 and provide a reference.
- Simplify and rewrite the sentence at line 132-133.
- Provide references for lines 155-164.
- Clarify the title of title 2 in Table 2.
- Add references for lines 186-192.
- Table 2: what should be the level of Acute kidney injury or Albuminurea during followup to suggest a biopsy in young age.Consider creating a separate heading to discuss co-morbidities and mortality in CKD.
- Correct the title in Table 3, title 2.
- Group SGLT2 inhibitors to enhance readability.
- In section 5.2.1, add a conclusion at the end based on points 6 and 7 regarding the use of ACEi/ARBs.
- Include statistics in the "Effect" columns in Table 4 if available.
- Create a table detailing available drugs and trials, including ongoing trials, related to GLP-1R/GIPR or combination with Glucagon agonists/SGLT2i.
Its okey, but could be improved. Some sentences are complex which could be simplified.
Author Response
Reviewer 1:
In the review titled "Chronic Kidney Disease in the Elderly Patient with Diabetes," the authors have summarized the current treatment options for CKD patients, both with and without diabetes mellitus (DM), and in different age groups. They discussed new therapies and optimal management for CKD patients, with a focus on elderly individuals. To enhance the review, consider the following suggestions:
- Introduction Clarity: The introduction should be more straightforward. In lines 32-37, specify the type of CKD being discussed, its differences in patients with diabetes (DKD) vs. normo-albuminuria, CKD in elderly vs. young patients, current therapies, their limitations, and the objectives of the review.
We would like to thank the reviewer for clarifying the scope of our review, and we revised the introduction to orient the reader to the content of our manuscript.
- This review would benefit from the inclusion of summary figures or flowcharts. For instance, Table 1 could be improved by incorporating histopathologic images or a model illustrating the pathologic classifications of DKD and highlighting the distinctions from non-DKD and aging, supported by the latest references.
Thank you for the suggestion to incorporate histopathological pictures. While we don’t have access to figures that we can incorporate in this publication, there are numerous open access (but not public domain) resources for such figures and point out the interested reader to these resources by providing the relevant references in the text. If the editor feel this is necessary, we can explore incorporating pictures from other journals.
- Staging Table: consider creating a table that stages CKD in different age groups to assist readers.
Thank you for this thoughtful suggestion we elected not use differences by age to remain aligned with the KDIGO guidelines. Nevertheless, the argument in support of age dependent cutoffs has been made rather forcefully in the literature. Hence, we included a new visualization based on the paper by Delanaye et al JASN 2019 30(10):1785-1805 and included this in the paper as Figure 1 (we opted for a figure rather than a table, because we could more easily incorporate colors in the former, rather than the latter as per the journal’s template).
Authors should consider discussing gender differences in CKD and their care management.
Thank you for the suggestion to include this information. However, the focus of our review is on the age differences (rather than the gender). None of the analyses reporting age effects ever reported age by gender race effects (a two-way classification). The lack of publicly available clinical trial data makes it impossible for us to comply with the great suggestion by the reviewer.
5. Treatment considerations: Authors should consider making a table – summarizing general approaches for treating CKD in elderly with or without DM, young people, with other comorbidities and include proper references. This would help separate clarity what authors are proposing later in this review as individual care managements
We would like to thank the reviewer for this suggestion. However, the material we have reviewed in this manuscript argue that there are no differences in the type of agents that should be used for the management of diabetic kidney disease between younger and older individuals with CKD and DM. Furthermore, this review focuses on the management of the elderly patient with CKD and DM, and many critical clinical trials to extrapolate in non-diabetic kidney disease (e.g. FIND-CKD for finerenone) are still ongoing. Therefore, the review can’t possibly address CKD in patients with and without DM due to the lack of data. We feel that a table would not add any information to this manuscript. Nevertheless, a strengthening of the recommendations at the conclusion is warranted, and hence we updated Section 6 to include:
“Hence, there should be no age discrimination in prescribing these newer agents for older individuals, as the trials to date suggest that older individuals will derive benefit from such therapies.”
- Pharmacological Interventions: Include a figure or model detailing various pharmacological interventions, their sites of action, and mechanisms.
We would like to thank the reviewer for the suggestion and we added a figure (Figure 3) to represent the site of action for the cardiorenal protective drugs in the field of diabetes and kidney disease.
DPP-4 Inhibitors: Discuss DPP-4 inhibitors, their mode of action, and relevant clinical trials.
We would like to thank the reviewer for the opportunity to consciously omit DPP4i from this review. None of the DPP4 agents have been shown to exert a renal protective effect and in fact the guidelines do not list them as priority drugs in the context of CKD with diabetes. In our review which focuses in the management of CKD in the context of diabetes, DPP4i are simply not relevant (though they are valuable agents for glycemic control, but without any obvious benefits for the cardiovascular and kidney complications. We thus politely decline to include them in the review, yet feel that this omission should be justified to the reader of this review. We thus included the following sentence in the introduction:
“In this review, we will focus predominantly on agents that exert cardiovascular and kidney benefits in individuals with CKD and DM, i.e. SGLT2i, GLP1RA and MRAs, omitting agents, such as the Dipeptidyl peptidase 4 inhibitors whose cardiovascular and kidney safety has been demonstrated in randomized clinical trials, but are currently not indicated to reduce the high cardiovascular and kidney risk in patients with DM and CKD.”
GLP1R/GIPR/Glucagon Triple Agonist: Consider updating the review with information on the ongoing study involving Retatrutide (LY3437943) or its potential effects on CKD. Additionally, discuss the potential benefits of a combination of GLP1R/GIPR agonists with SGLT2 inhibitors (e.g., Tirzepatide). Please review and incorporate following references,
Clinical trials : NCT05936151; NCT05536804;
Irene Caruso et al. eClinicalMedicine. 2023 Oct; 64: 102181.
Ania M. Jastreboff et al. N Engl J Med 2022; 387:205-216
Shweta Urva et al. The Lancet, Volume 400, Issue 10366,2022, Pages 1869-1881,
We would like to thank the reviewer for pointing out these studies and the additional references for the peptide section. We have incorporated these references and the clinical trial registration number. The reference by Caruso et al is used to justify the combined use of SGLT2i and GLP, a point made in a recent paper in Circulation (Circulation. 2023 Nov 12. doi: 10.1161/CIRCULATIONAHA.123.067584.) by Neuen et al. The latter were added to the conclusion section of the paper.- “… and propose a framework for the management of DKD…” this proposal is missing at the end.
We would like to thank the reviewer for pointing out this non sequitur. The section was rephrased accordingly.
Minor Corrections:
- Rewrite the sentence at line 63-64 and provide a reference.
We believe that the entire section between lines 63-68 in the original version of the document suffered from an unlucky combination of “accept changes” and copyediting. Hence, we rewrote the section and corrected some of the percentages.
- Simplify and rewrite the sentence at line 132-133.
- Provide references for lines 155-164.
- Clarify the title of title 2 in Table 2.
- Add references for lines 186-192.
Thank you for the suggestions 2-3, which we addressed in the revision. We also took the liberty to rename the Table concerning the indications for a kidney biopsy.
- Table 2: what should be the level of Acute kidney injury or Albuminurea during follow-up to suggest a biopsy in young age. Consider creating a separate heading to discuss co-morbidities and mortality in CKD.
We would like to thank the reviewer for the opportunity to clarify this point. There is no specific eGFR at which a biopsy is indicated (or contra-indicated, though advanced CKD with small, scarred kidneys on imaging are considered a contraindication). We added a clarifying statement about the rationale behind some of the criteria offered in the literature.
- Correct the title in Table 3, title 2.
Please see our response to reviewer 2 – this was a typo which we addressed.
- Group SGLT2 inhibitors to enhance readability.
Thank you for pointing this imperfect visualization to us. The SGLT2i meta-analysis figure was regrouped hierarchically: by drug, trial, outcome, and patient group for easier comparisons.
- In section 5.2.1, add a conclusion at the end based on points 6 and 7 regarding the use of ACEi/ARBs.
This statement was added and a reference was provided.
- Include statistics in the "Effect" columns in Table 4 if available.
Thank you for making this suggestion Unfortunately the p-values were not provided in the source publications. We computed them based on simple Monte Carlo calculations and verified that the differences were not statistically significant and made such a statement in the text.
- Create a table detailing available drugs and trials, including ongoing trials, related to GLP-1R/GIPR or combination with Glucagon agonists/SGLT2i.
Thank you for the suggestion which we cannot accommodate in its entirety due to lack of space (a cursory examination of clinicaltrials.gov revealed a double digit of trials that refer to the combination, many of which are still recruiting patients). Reviewer 2 suggested two clinical trial registrations to report in the text (which we did). Furthermore, we cite two targeted meta-analysis (the paper by Caruso et al EClinical Medicine 2023 & Li et al Frontiers Pharmacology 2022) about the combination in the conclusion section of the manuscript. This should provide sufficient information to the interested reader to follow up the literature, including the actuarial analysis by Neuen et al published at Circulation in late November 2023 (this paper is also cited in the conclusion section).

Reviewer 2 Report
Comments and Suggestions for Authors
Chronic Kidney Disease in the Elderly Patient with Diabetes
Ravender R et al.
Journal of Clinical Medicine
After reviewing the evidence, Authors concluded that the benefits of the emerging pharmacological standard of care for Diabetic Kidney Disease (DKD) in terms of reduction of cardiovascular and kidney outcomes are similar in elderly and non-elderly subjects with type 2 diabetes. Authors suggest that, for many aspects, care should be applied quite equally in non-elderly and elderly individuals with type 2 diabetes and DKD.
The following observations are reported as encountered in the text:
1. Row 33: “often”, we suggest changing in “almost systematically.”
2. Row 34: “Diabetes in CKD”; the term “Diabetes in CKD” is attributed to the KDIGO 2022 Guidelines, but that definition does not appear in that document. Really, I think that it is not necessary to introduce this new definition to describe the complex and often multifactorial pathogenesis of kidney disease in individuals with diabetes that is already included in the DKD term. Indeed, DKD also includes the increasingly recognized non-albuminuric DKD phenotype of kidney disease in subjects with diabetes. Thus, we suggest rephrasing rows 32.37 accordingly. Consistently, I suggest using “DKD” in place of “DM and CKD” where applicable throughout the manuscript (for instance, row 38, row 54, row 90, row 93, row 96 …., row 229, row 232, and so on).
3. Row 64: “diabetes” in place of “diabete”
4. Rows 66-68: the sentence is unclear or pleonastic. I suggest starting with sentence with “ESRD among patients with type 2 DM (3) is considered ….”
5. Rows 76-79: this sentence too, appears unclear. Please, rephase. Do the percentages reported refer to the diabetic population or to the general population? Please, add the relevant reference.
6. Section 2: interesting and updated information about the incidence of CKD among adults with diabetes have been recently published by the New Engl J Med (387: 1430-1431, 2022; Tuttle KR et al.). That paper reports data for elderly subjects (even those with ≥80 age years) and different races and ethnicities. This paper deserves consideration.
7. Row 113; “RAGR”, likely “RAGE”,
8. Row 128: “RAS”, this abbreviation is useless and misleading. Please, avoid it.
9. Rows 135-138: this sentence is unclear; please rephrase. “microalbumin to creatinine ratio”?
10. Row 231: “this is integrative”; please, correct.
11. Row 266: “eldelry” is “elderly”.
12. Row 269: “IDT” really is “IDNT”.
13. Figure 1: the figure looks incomplete. Diamonds do not appear in the figure together with data about the meta-analysis for a random effects model. Furthermore, in the line assigned to the DELIVER study, >=72 to <72.
14. Table 3: what “Title 2” means?
15. Row 382: 95% CI really is “-8.22 to -1.75 mmHg”; furthermore, for protein excretion use the same unit of measure (mg or g).
16. Row 444: 95% CI really is 0.80 – 0.93; p<0.0001 (as reported in reference 139; Sattar N et al). Please, use the correct data.
17. You have described the two composite kidney outcomes employed in the meta-analysis published by Sattar N et al. I therefore suggest briefly describing the results that emerge from this meta-analysis in relation to both renal outcomes even if age-stratified data are not available, as it is for MACE.

I suggest a review of the paper by a native English speaker
Author Response
Reviewer 2
- Row 33: “often”, we suggest changing in “almost systematically.”
This change was made in the text.
Row 34: “Diabetes in CKD”; the term “Diabetes in CKD” is attributed to the KDIGO 2022 Guidelines, but that definition does not appear in that document. Really, I think that it is not necessary to introduce this new definition to describe the complex and often multifactorial pathogenesis of kidney disease in individuals with diabetes that is already included in the DKD term. Indeed, DKD also includes the increasingly recognized non-albuminuric DKD phenotype of kidney disease in subjects with diabetes. Thus, we suggest rephrasing rows 32.37 accordingly. Consistently, I suggest using “DKD” in place of “DM and CKD” where applicable throughout the manuscript (for instance, row 38, row 54, row 90, row 93, row 96 …., row 229, row 232, and so on).
We would like to thank the reviewer for pointing out this mis-attribution (which we corrected). The term CKD in diabetes will be used increasingly in place of DKD, (e.g. even the American Society of Nephrology Diabetic Kidney Disease Consortium committee of which one of the authors is a member) is shifting terminology because of the multifactorial nature of the disease, and the clinical rather than histological diagnosis. Furthermore, none of the clinical trials reviewed required a histological diagnosis to recruit patients in the trials, i.e. participants formally had CKD with diabetes type 2, rather than DKD. Hence, we feel it is important that the newer terminology is retained, and its relationship with the older one clarified. In the revision, we opted to retain both terms and clarify the rationale in the introduction. We do agree with the reviewer that the term DKD is underused and we made a few of the replacements in the places suggested.
- Row 64: “diabetes” in place of “diabete”
Thank you for noting this error. This sentence was deleted in the revision due to the repetitive nature of the content.
- Rows 66-68: the sentence is unclear or pleonastic. I suggest starting with sentence with “ESRD among patients with type 2 DM (3) is considered ….”
We merged this sentence with the information in the presenting paragraph to provide a clearer link between obesity, diabetes and CKD. See also response to Reviewer 1 (under minor comment 1).
- Rows 76-79: this sentence too, appears unclear. Please, rephrase. Do the percentages reported refer to the diabetic population or to the general population? Please, add the relevant reference.
Thank you for pointing out the lack of clarity in the sentence. The percentages are about patients with diabetes and the reference (Afkarian et al JAMA 2016, 316, 602–610, doi:10.1001/jama.2016.10924) was added to the text.
- Section 2: interesting and updated information about the incidence of CKD among adults with diabetes have been recently published by the New Engl J Med (387: 1430-1431, 2022; Tuttle KR et al.). That paper reports data for elderly subjects (even those with ≥80 age years) and different races and ethnicities. This paper deserves consideration.
Thank you for this great citation, which we incorporated in the text.
- Row 113; “RAGR”, likely “RAGE”,
Thank you for pointing out the typo, which we corrected in the revision
- Row 128: “RAS”, this abbreviation is useless and misleading. Please, avoid it.
Thank you for making this great suggestion; we removed the abbreviation from the text.
- Rows 135-138: this sentence is unclear; please rephrase. “microalbumin to creatinine ratio”?
Rewrote to convey the meaning of the urinary albumin to creatinine ratio in a spot urine sample.
- Row 231: “this is integrative”; please, correct.
Corrected the typo.
- Row 266: “eldelry” is “elderly”.
Corrected the typo.
- Row 269: “IDT” really is “IDNT”.
Corrected the typo.
- Figure 1: the figure looks incomplete. Diamonds do not appear in the figure together with data about the meta-analysis for a random effects model. Furthermore, in the line assigned to the DELIVER study, >=72 to <72.
Thank you for giving the opportunity to clarify. The diamonds are “adjusted” predictions, that consider various factors across all the trials, while the whiskers plot are the observed hazard ratios. Hence the difference between the diamonds and the whisker plots are the usual differences between adjusted and non-adjusted analyses. In the revision we corrected the typo, changed the shading (which made it difficult to see the diamonds), and rearranged the plot, so that the estimates for the young and old subgroups from the same study appear in consecutive “rows” of the figure to make the comparisons somewhat easier to appreciate.
- Table 3: what “Title 2” means?
Thank you for pointing this out. Title 2 was left from the excel version of the table. It was meant to stand for “At risk”, and the third column header was supposed to read as “Preventive Measures”. These typos were corrected in the revised version. Note that the original Table 3, is now Table 4.
- Row 382: 95% CI really is “-8.22 to -1.75 mmHg”; furthermore, for protein excretion use the same unit of measure (mg or g).
- Row 444: 95% CI really is 0.80 – 0.93; p<0.0001 (as reported in reference 139; Sattar N et al). Please, use the correct data.
We appreciate the reviewer catching these two typos, which we corrected in the revision.
- You have described the two composite kidney outcomes employed in the meta-analysis published by Sattar N et al. I therefore suggest briefly describing the results that emerge from this meta-analysis in relation to both renal outcomes even if age-stratified data are not available, as it is for MACE.
We concur with the reviewer that this is important background information which we provided in the revision.
Reviewer 3 Report
Comments and Suggestions for Authors
The manuscript <Chronic Kidney Disease in the Elderly Patient with Diabetes> by Ravender R et al reviews the evidence of benefits from newer therapies applied to elderly and young patients with diabetes mellitus.
I expected to read why age is a limiting factor in a new therapy, what are the benefits in relation to age what are the complications, and why the drug SGLT2 is not indicated for type 1 diabetes.
I'm sorry, I'm not satisfied.
Author Response
Reviewer 3
The manuscript <Chronic Kidney Disease in the Elderly Patient with Diabetes> by Ravender R et al reviews the evidence of benefits from newer therapies applied to elderly and young patients with diabetes mellitus.
I expected to read why age is a limiting factor in a new therapy, what are the benefits in relation to age what are the complications, and why the drug SGLT2 is not indicated for type 1 diabetes.
I'm sorry, I'm not satisfied.
In our paper we examined systematically the lack of any age limitations with regards to these therapies. It was precisely our point to illustrate through the review of the pathophysiology and the clinical trial evidence that that there are no age-related limitations (the drugs work equally well in young and old) and no apparent safety signals. We limited our attention to agents that one can use today across locales (e.g. SGLT2i are indicated across the globe for patients with Type 2 diabetes and CKD for their cardiovascular and kidney benefits), and with the exception of Dapagliflozin in EMA, where the drug is indicated as an adjunct for glycemic control, they are not indicated anywhere else. In fact, the diabetic ketoacidosis will likely be a major limiting factor if these agents ever get approval for kidney and heart protection in the US (or Japan). We can’t speculate what this indication will look like, hence we elected not to cover type 1 diabetes at all, despite the clear need of further therapies for kidney disease for these patients.
Round 2
Reviewer 1 Report
Comments and Suggestions for Authors
The article has undergone significant improvement. There is still some room for improvement of Figure 3. For instance, including the names of the most common drugs in each section would make the figure more informative.
I have no further comments.
Author Response
Thank you for the comments. We included the trade name of representative members of the class in the figure legend.
Reviewer 2 Report
Comments and Suggestions for Authors
I thank the authors for accepting the comments and for their replies.
In the text, page 4, row 308, Figure 3 is actually Figure 2. Please, correct in the text and in the legend of the Figure.
Author Response
Thank you for pointing out the reference to the wrong image. This was deleted in the revision.